# Design and Experiment of Greenhouse Self-Balancing Mobile Robot Based on PR Joint Sensor

**Yaohui Zhang [1], Yugang Song [1], Fanggang Lu [1], Dongxing Zhang [1,2], Li Yang [1,2], Tao Cui [1,2], Xiantao He [1,2] and Kailiang Zhang [1,2,*]**

[1]  College of Engineering, China Agricultural University, Beijing 100083, China
[2]  Key Laboratory of Soil-Machine-Plant System Technology of Ministry of Agriculture, Beijing 100083, China
*   Correspondence: zhang_kailiang@cau.edu.cn; Tel.: +86-010-62737765

**Abstract:** To avoid issues such as the greenhouse working robot's inability to perform normal tasks or reduced working accuracy due to the influence of uneven ground, this study designed a set of greenhouse self-balancing mobile robots. The self-balancing mobile robot system designed in this study uses a quadruped mobile robot as a carrier, equipped with a three-degrees-of-freedom wheel-leg structure and is complemented with a posture control algorithm. The algorithm calculates the adjustment of each leg based on the vehicle's tilt angle and wheel-ground pressure, achieving control over the robot's posture angle, the center of gravity height, wheel-ground contact force, and other functions. To address the issue of over-constrained (weak legs) posture adjustment during mobile robot fieldwork, a flexible joint sensor based on the PR structure has been designed and developed. After field testing, it was verified that the greenhouse self-balancing mobile robot proposed in this study can adapt well to field environments, such as climbing hills, overcoming obstacles, crossing furrows, and so on. The response speed of the flexible joint sensor can meet the requirements of self-balancing while effectively solving the problem of weak legs.

**Keywords:** mobile robot; self-balancing; displacement amplification structure; pose adjustment system; greenhouse robot

## 1. Introduction

Due to the influence of uneven ground, the harvesting efficiency of greenhouse-picking robots can be greatly affected [1,2], and losses may also occur during the harvesting and transportation process due to ground undulations [3]. Therefore, greenhouse mobile robots need to have high posture stability and obstacle-crossing capabilities to meet the needs of greenhouse harvesting and transportation robots [4–6]. Therefore, the design and development of a set of greenhouse self-balancing mobile robots suitable for agricultural harvesting environments can not only improve the application and promotion of greenhouse-picking robots but also provide a reliable solution for the transportation of greenhouse products.

To adapt to different working environments, domestic and foreign researchers have developed numerous mobile robots [7,8]. Among them, the most important research and development goal is to combine the fast movement ability of wheeled robots with the terrain adaptation and posture control ability of legged robots [9,10].

Fan Guiju et al. from Shandong Agricultural University designed a self-balancing control system for a fruit orchard lifting platform. The system uses an inclination sensor installed on the working platform to monitor the tilt degree in real time and adjusts the working platform based on the angle data, with a maximum leveling error of 1.74° [11]. Lin Chen et al. addressed the precision weeding requirement of a weeding robot by using a gyroscope to collect and process the biaxial angle, and by using a combination of a geomagnetic sensor, an accelerometer, and other sensors to level the operating platform, they enabled the control system to achieve precise control of the weeding device end [12].

Compared to the scholars mentioned in the following text, the research mentioned above primarily focuses on the self-balancing of the operating platform rather than the mobile platform. This is one of the methods to achieve self-balancing of the operating platform; however, it has limitations. The scholars mentioned in the following text, on the other hand, focus on the self-balancing of the mobile platform by adjusting its posture to adapt to complex environments.

A mobile robot equipped with an active articulated suspension system was designed to adapt to motion in rugged terrain; however, it can only adjust the pitch angle and did not consider the problem of poor wheel contact due to rough terrain [13]. The posture and stability control model for a wheeled-legged mobile robot in rugged terrain can estimate the normal contact force to maintain continuous contact between the wheels and the ground, resulting in an improvement in poor wheel–ground contact [14]. An actively articulated suspension (AAS) reconfiguration method is proposed for a robotic vehicle with AAS to negotiate an obstacle during straight motion. To address the obstacle avoidance problem during linear motion, an active articulated suspension was designed, which enables the robot to maintain a minimum static stability of over 0.6 radians. However, the design of this robot ignores lateral and diagonal movements [15]. William Reid and others proposed a balance-control method based on a kinematic model and an online-generated terrain map; however, the method relies on an RGB-D sensor to collect three-dimensional point cloud information [16]. Qi Wenchao and his colleagues proposed a double closed-loop fuzzy PID control method for the body leveling of tractors in hilly terrain under complex working conditions. The method achieved a static leveling time of 12.5 s and a leveling error control within 0.5° when the inclination angle was 15° [17]. The team of Yin Xiang designed a self-balancing system based on electro-hydraulic control to solve the problem of lateral inclination of high ground-clearance sprayers during operation. The experimental results showed that the maximum leveling error was 1.53° and the root mean square error was 0.454° [18]. Florian Cordes et al. studied the mechatronic design and control methods of the hybrid wheeled-legged exploration vehicle SherpaTT [19]. The active suspension system takes the pitch and roll data of the vehicle body, as well as the six-axis force–torque sensor data at the end of the wheel-legs, as inputs to achieve attitude control in unstructured terrain. However, this solution has a complex robot structure and high cost and is not suitable for agricultural production. The above scholars mainly use six-axis force sensors or read the feedback current of the leveling motor to obtain foot pressure information when solving the virtual leg problem of leveling. The main problem of using a six-axis force sensor to solve the virtual leg problem is its high cost, which is not suitable for the agricultural field or micro mobile platforms [14,19]. The main problem with reading the feedback current of the leveling motor is that the foot pressure passes through the mechanical transmission equipment, resulting in inaccurate detection values of the motor current [20,21].

The above-mentioned wheeled-legged mobile robots have good adaptability to complex terrain. Their multi-degree-of-freedom leg design allows the robot to fully adjust its posture according to changes in the terrain and can achieve higher maneuverability by adjusting its gait when encountering obstacles or gullies. However, the design of multi-joint and multi-degree-of-freedom legs leads to higher structural complexity, complex control models, and increased manufacturing costs. According to research by domestic and foreign researchers in the field of mobile robots, many mobile robots have been studied for various applications. There are also corresponding solutions to the "weak leg" problem of mobile robots; however, their solutions are relatively costly or are not suitable for small environments such as greenhouses. The term "weak leg" refers to the issue where, in a four-wheel robot achieving balance, one or two wheels may have significantly less contact force with the ground compared to the other three wheels. In this situation, the stable state of the robot platform is easily disrupted, which can have an impact on other operational equipment. This study proposes a PR flexible joint sensor for detecting foot pressure, which has a lower cost and is suitable for the legs of small-scale robots while ensuring leg rigidity.

In conclusion, this study aims to develop a set of self-balancing mobile robots suitable for narrow greenhouse environments, providing a stable and adaptable mobile platform for greenhouse harvesting and transportation robots to operate in rough and narrow environments. The main research content of this study on self-balancing mobile robots includes (1) developing and designing a three-degrees-of-freedom wheeled-leg robot suitable for narrow greenhouse environments; (2) developing a self-balancing control system that can dynamically adjust the robot's lateral and longitudinal tilt angles; and (3) to solve the "weak legs" problem, we designed a flexible joint sensor based on the PR joint.

## 2. Materials and Methods

Self-balancing mobile robots can solve various complex application scenarios in greenhouse environments, providing a reliable mobile platform for the precise and targeted application of fertilizers and pesticides, harvesting, and transportation, reducing the difficulty and promoting the application of precision agriculture technologies. Based on the analysis of the limitations and requirements of greenhouse environments, this study developed a self-balancing mobile robot that meets specific work requirements with flexibility, stability, and self-balancing ability. In this section, we will provide a detailed introduction to the design of the greenhouse self-balancing mobile robot developed in this study.

### 2.1. Design Scheme and Working Principle of Self-Balancing Mobile Robot in Greenhouse

The four-wheel layout can balance flexibility and stability; therefore, this paper designed a self-balancing mobile robot for facility harvesting based on the four-wheel layout, which is mainly used for facility agriculture harvesting and transportation. The structure of the facility harvesting self-balancing mobile robot is shown in Figure 1. The greenhouse self-balancing mobile robot mainly consists of a self-balancing control system, a three-degrees-of-freedom wheeled-leg system, and a walking control system. The three-degrees-of-freedom wheel-leg is primarily responsible for the robot's walking mode and robot posture control. The casings of the wheel-leg, flexible joint sensor, and wheel-leg drive unit are mainly designed to protect the motors and sensors from damage.

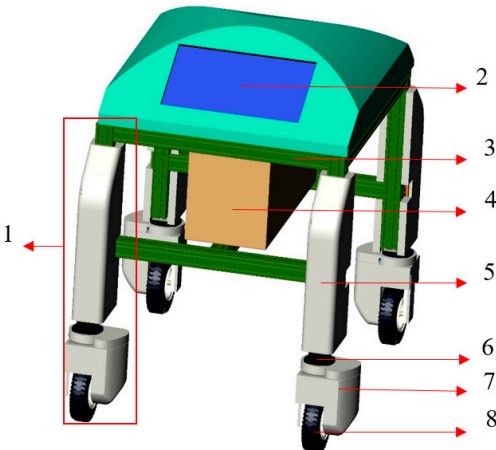

**Figure 1.** Design of greenhouse self-balancing mobile robot. Note: components, No.: 1. three-degrees-of-freedom wheel-leg; 2. human–machine interface; 3. robot frame; 4. control cabinet; 5. wheel-leg shell; 6. flexible joint sensor shell; 7. wheel-leg drive unit shell; 8. wheel hub.

The three degrees of freedom of the three-degrees-of-freedom wheeled-leg system are responsible for driving, steering, and balancing, as shown in Figure 2a. A single three-degrees-of-freedom wheeled-leg system has three drive motors, which are the leveling control motor(china), steering motor(china), and walking motor(china). The walking control system is designed with walking and steering modes. By controlling the motion of the three-degrees-of-freedom wheeled-leg system, the robot can achieve in-place turning, Ackermann steering, lateral movement, and diagonal 45-degree walking. The sensor is

installed between the wheel foot and the leg. The flexible joint sensor(china) is connected to both the wheel foot and the leg using bolts, as shown in Figure 2b.

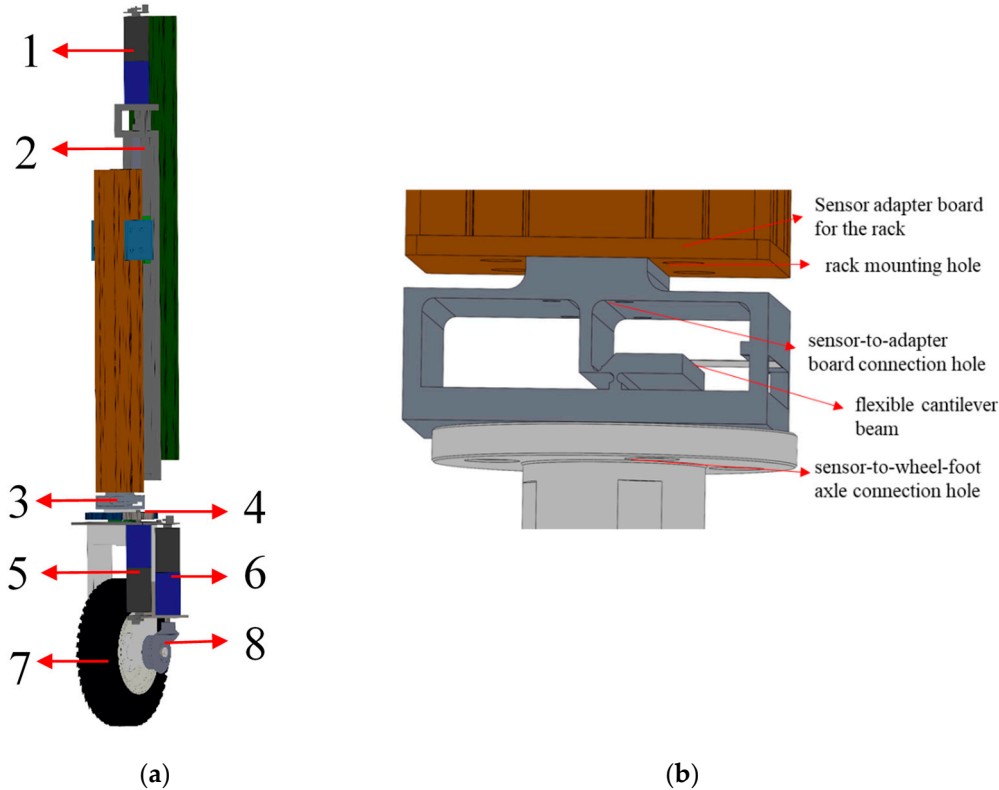

(**a**)                                                        (**b**)

**Figure 2.** (**a**) Design of the three-degrees-of-freedom wheeled-leg system. Note: components, No.: 1. leveling control motor; 2. leveling screw module; 3. flexible joint sensor; 4. steering gear set; 5. steering motor; 6. walking motor; 7. wheel hub; 8. steering bevel gear set. (**b**) Partial view of the sensor.

The self-balancing control system mainly ensures that the actuator mounting platform of the robot remains in a horizontal position during walking or stationary states. The self-balancing control system can provide a stable relative reference position for the working device, avoid platform tilting, and prevent errors in the working coordinate system of the working manipulator or material spillage during transportation due to tilting. The attitude control of the robot mainly relies on the reciprocating linear motion of the telescoping joint to perform its work. The telescoping joint is powered by a DC servo motor(china), which drives a screw on the fixed leg to rotate, thereby driving the telescoping leg to perform reciprocating linear motion and complete the leg extension and retraction. Attitude control requires vehicle tilt angle data as input; therefore, the self-balancing mobile robot integrates a gyroscope to obtain the pitch angle, roll angle, and angular velocity data of the vehicle. However, to solve the problem of poor contact between the legs and the ground during the self-balancing process (the "weak leg" problem), it is necessary to obtain pressure data from the robot's wheels and the ground. There are mainly two ways to obtain force data at the end of the mobile robot's wheels or legs. One is to indirectly calculate the end force through the motor's torque feedback, and the other is to directly measure the end force by adding force sensors [14,22]. Due to the significant mechanical losses in the telescoping joint caused by the screw transmission, the motor torque feedback cannot accurately reflect the pressure at the end of the wheeled leg. Therefore, in this study, a PR flexible joint sensor was designed to detect leg pressure. To ensure the rigidity of the self-balancing mobile robot and the accuracy of the foot pressure detection, the PR flexible sensor is installed at the connection between the wheeled foot and the telescoping joint. The full-bridge strain gauges(china) are attached to a flexible cantilever beam to detect the contact force

between the foot and the ground, as shown in Figure 2b. The variation in foot pressure causes slight displacement at the junction plate, which is amplified by the PR structure and converted into an electrical signal by the strain gauges, which is then transmitted to the robot's self-balancing system. The self-balancing system of the robot can determine the magnitude of the foot contact force based on the information provided by the PR flexible joint sensor. By integrating the information from the PR flexible sensor and the tilt angle sensor, the robot adjusts its posture according to the vehicle's posture control model. To protect the internal sensors from water and other damage in the greenhouse environment, the shell of the wheeled foot and leg, the shell of the flexible joint sensor, and the shell of the wheeled foot drive unit are designed to protect the sensors.

### 2.2. Research on Control Method of Body Attitude Angle

Controlling the vehicle's attitude angle is an effective method for maintaining the stability of the robot's posture. However, due to the complex and unknown nature of the terrain, in addition to the vehicle's attitude angle, attitude control also needs to consider the following factors. The first factor to consider is the change in the center of gravity height caused by terrain changes and accumulated errors during robot motion. Maintaining the stability of the center of gravity height is beneficial to enhance driving stability and avoid issues where the adjustment mechanism exceeds its travel range due to the continuous rise or fall of the center of gravity.

The second factor is the speed control of the adjustment. To reduce the system lag caused by untimely adjustments and vehicle body vibration caused by overshoot, the telescoping speed of each leg needs to adapt to the terrain changes.

#### 2.2.1. Attitude Angle Control Model Establishment

This section introduces the attitude angle control method based on ideal ground conditions (i.e., an ideal plane with a hill). Before introducing the control method, several related concepts are defined, as shown in Figure 3:

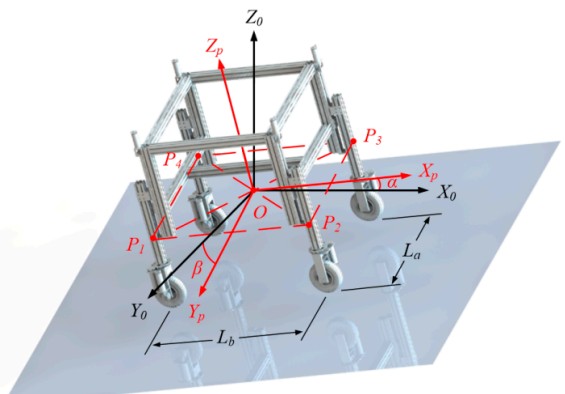

**Figure 3.** Simplified coordinate transformation model for a mobile robot.

Reference point: The midpoint of the telescoping joint stroke of each leg (fixed point, not moving with the telescoping leg), denoted as $Pi$.

Vehicle plane: The plane formed by the four reference points, denoted as $P_1P_2P_3P_4$.

Vehicle coordinate system: The origin is set at the geometric center point O of the vehicle plane. The $Zp$-axis is perpendicular to the vehicle plane, and the $Y_p$-axis is parallel to the line passing through $P_1$ and $P_4$, denoted as $\Phi_p$.

Horizontal coordinate system: The origin is set at the geometric center point O of the vehicle plane. The $Z_0$-axis is perpendicular to the horizontal plane, and the $Y_0$-axis is the projection of the $Yp$-axis onto the horizontal plane, denoted as $\Phi_0$.

This article mainly considers the need for the vehicle to maintain a horizontal posture when walking in a greenhouse; therefore, the control targets for pitch and roll angles are

set to 0. To achieve the fastest response speed, the attitude angle is adjusted based on the principle of keeping the center point stationary. That is, in each adjustment cycle, the four legs are synchronized to extend or retract based on the geometric center point of the vehicle plane. Based on this adjustment principle, the telescoping amount of each leg is calculated [23,24].

Assuming that the coordinates of any point P on the body in $\phi_p$ are given by $(i, j, k)$, we want to perform a coordinate transformation to obtain its coordinates in $\phi_0$, denoted as $(i_0, j_0, k_0)$.

First, we keep the coordinate $\beta$ unchanged and rotate $\phi_p$ around the $Y_P$-axis until $\alpha = 0$. The rotation matrix at this point is:

$$R_y(\alpha) = \begin{bmatrix} \cos\alpha & 0 & \sin\alpha \\ 0 & 1 & 0 \\ -\sin\alpha & 0 & \cos\alpha \end{bmatrix} \tag{1}$$

Next, we rotate $\phi_p$ around the $X_P$-axis until $\beta = 0$. The rotation matrix at this point is:

$$R_x(\beta) = \begin{bmatrix} 1 & 0 & 0 \\ 0 & \cos\beta & \sin\beta \\ 0 & -\sin\beta & \cos\beta \end{bmatrix} \tag{2}$$

The rotation matrix from $\phi_p$ to $\phi_0$ is:

$$ {}_0^H R = R_y(\alpha) R_x(\beta) = \begin{bmatrix} \cos\alpha & 0 & \sin\alpha \\ -\sin\alpha\,\sin\beta & \cos\beta & \sin\beta\cos\alpha \\ -\cos\beta\,\sin\alpha & -\sin\beta & \cos\beta\cos\alpha \end{bmatrix} \tag{3}$$

Due to the small changes in inclination angle during the adjustment process, higher-order terms can be ignored, and we can make the following approximation:

$$\cos\alpha = \cos\beta = 1, \sin\alpha = \alpha, \sin\beta = \beta \tag{4}$$

Therefore, the following formula can be obtained:

$$ {}_0^H R \begin{bmatrix} 1 & 0 & \alpha \\ 0 & 1 & \beta \\ -\alpha & -\beta & 1 \end{bmatrix} \tag{5}$$

The coordinates of point P under $\phi_0$ are deduced as:

$$(i_0, j_0, k_0)^T = {}_0^H R (i, j, k)^T \tag{6}$$

Let the coordinates of point Pi in $\phi_p$ be $(x_i, y_i, 0)$, and the vertical coordinate of Pi in $\phi_0$ be:

$$Z_{0i} = -\alpha x_i - \beta y_i \tag{7}$$

Then, the height difference between point Pi and the geometric center point O is:

$$e_i = z_{0i} - z_{00} = -\alpha x_i - \beta y_i \tag{8}$$

Let $e_i$ be the telescopic leg extension amount. It should be noted that due to the inclination angle between the telescopic leg and the horizontal plane in the initial state, when the telescopic leg extends by $e_i$, the geometric center of the vehicle body plane will undergo a slight downward shift; however, it will not affect the accuracy of attitude angle adjustment.

2.2.2. Adjusting Speed Control Model

To achieve high-precision dynamic adjustment, it is necessary to control the extension and retraction speed of each wheel-leg. After obtaining the adjustment amount, if each wheel-leg is adjusted at a constant speed, when the attitude angle is adjusted to a small range, the extension and retraction joints will adjust the tiny angle at a relatively high speed, which is prone to over-adjustment, causing the inclination angle to repeatedly adjust around the desired angle. When the robot moves on complex terrain, the body will continue to shake [25,26].

Therefore, a PID controller is introduced for adjustment. The expression for the proportional adjustment term is:

$$v_{pi}(t) = K_p \varepsilon(t) \tag{9}$$

In the equation, $v_i(t)$ represents the speed of any telescopic leg at time $t$, $\varepsilon(t)$ represents the inclination angle data at time $t$, and $K_P$ represents the proportional coefficient. Since the inclination angle data contain pitch and roll angles, and in Section 2.2.1, we have obtained the relationship between the adjustment amount of each telescopic leg under the principle of center point immobility and the two sets of inclination angles; therefore, it is possible to indirectly achieve a proportional adjustment of the inclination angle to the adjustment speed by establishing a proportional relationship between the adjustment speed and the adjustment amount. The expression is:

$$v_{pi}(t) = K_p e_i(t) \tag{10}$$

The expression of the integral adjustment term is:

$$v_{Ii}(t) = K_I \int_0^t e_i(t)dt \tag{11}$$

The expression of the differential adjustment term is:

$$v_{Di}(t) = -K_D \frac{de_i(t)}{dt} \tag{12}$$

From Formula (8):

$$\frac{de_i(t)}{dt} = -x_i \frac{d\alpha}{dt} - y_i \frac{d\beta}{dt} \tag{13}$$

Let the angular velocities of pitch angle $\beta$ and roll angle $\alpha$ be $\omega_\beta$ and $\omega_\alpha$, then the differential adjustment term can be expressed as:

$$v_{Di}(t) = K_D(x_i \omega_\alpha + y_i \omega_\beta) \tag{14}$$

Among them, $\omega_\beta$ and $\omega_\alpha$ can be obtained directly by the gyroscope.

In summary, the speed expression of each wheel-leg under PID regulation is as follows:

$$v_i(t) K_P e_i(t) + K_I \int_0^t e_i(t)dt + K_D(x_i \omega_\alpha + y_i \omega_\beta) \tag{15}$$

In the formula, $K_P$, $K_I$, and $K_D$ are all greater than zero.

According to the above attitude angle control method, it is theoretically possible to achieve a self-balancing adjustment of the robot. However, for a four-wheel layout robot, even if one of the four wheels has poor or no contact with the ground, the robot can still maintain a horizontal state. This phenomenon is also known as the "weak leg" phenomenon and belongs to a hyperstatic problem. In this case, the robot may rapidly and significantly tilt, which can be more harmful. Therefore, this study designed a PR flexible joint sensor to address the "weak leg" problem that self-balancing robots may face.

*2.3. The Design of PR Flexible Joint Sensor*

Section 2.2 provides a detailed description of the body attitude angle control method used in this study. However, this method cannot solve the "weak leg" problem. Therefore, in this study, a PR flexible joint sensor was designed to detect the contact force between the wheels and the ground, which was used to solve the "weak leg" problem.

2.3.1. Constraint Analysis of Flexible Joint Sensor

The support leg is the main structure that ensures the stability of the system and bears the load. It also performs self-balancing of the platform through its extension and retraction motion. The mechanical system adopts an aluminum profile with a planetary reducer and a DC motor to drive the trapezoidal screw, ensuring the rigidity and complete self-locking of the support leg, as shown in Figure 2a. The support leg is arranged vertically and driven by a DC motor (Figure 2a (1)) to rotate the trapezoidal screw (Figure 2a (2)), which in turn drives the sliding block and the telescopic leg connected to it to perform extension and retraction motion.

Since the support-leg module and the wheel module are independent of each other, the designed flexible joint is installed between them and connected to both modules. This will not only not affect the rigidity of each independent module but will also allow the foot pressure to act directly on the flexible joint. As the flexible joint is directly connected to the support-leg module, its rigidity design must take into account the impact of the maximum pushing force of the support leg. According to the formula for support-leg pushing force [27],

$$\mathrm{F} = \frac{2\pi i \mathrm{T}\eta}{s} \tag{16}$$

In the equation, $i$ is the mechanical transmission ratio, which is 3.7;

T is the rated torque of the motor, which is 1.2 N·m;

$\eta$ is the mechanical efficiency, where the transmission efficiency of the universal coupling is 0.95 and that of the trapezoidal screw is 0.26; therefore, the mechanical efficiency of the support leg is 0.247;

s is the lead of the trapezoidal screw, which is 10 mm.

By substituting the above data into the formula, the pushing force of the support leg is calculated as 689 N. When designing the flexible joint, the rigidity of the joint must be maintained well under the extreme pushing force.

For mobile robots, conventional, simple motion pairs may not be suitable due to the requirements for rigidity and installation space. In the joint configuration of a robot, using flexible motion pairs instead of actual motion pairs allows micro-robots to not only eliminate the clearance, friction, and backlash associated with conventional motion pairs but also exhibit high rigidity, high precision, high speed, and other characteristics.

Therefore, this study proposes a flexible joint structure based on a lever-type lever-cantilever double-stage micro-displacement amplification structure that uses flexible motion pairs, as shown in Figure 4a. It mainly consists of a PR-type displacement transfer amplification structure and a pressure signal output structure. After being connected to the end of the original platform support leg, the support-leg ground contact pressure is converted into displacement and transferred to the pressure signal output structure through the PR-type displacement amplification structure, realizing real-time detection of foot pressure. The overall basic size of the flexible joint is designed according to the size of the support-leg structure, which ensures the reduction of the impact on leg stiffness while minimizing the increase in support-leg length. The detailed structure of the PR flexible joint sensor is shown in Figure 4b.

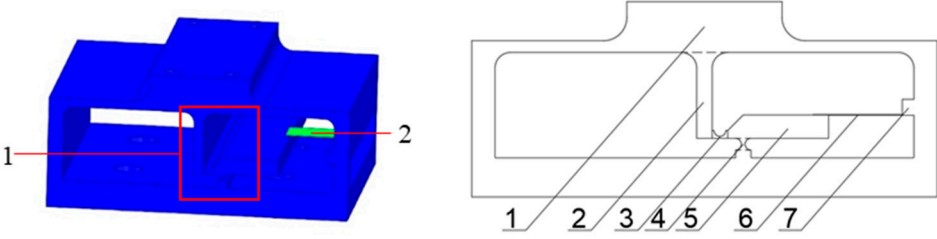

(**a**) PR flexible joint 3D design drawing  (**b**) PR flexible joint structural representation

**Figure 4.** PR flexible joint structure. Note: components (**a**) PR flexible joint 3D design drawing, No.: 1. displacement signal output structure; 2. PR-type displacement transfer; and (**b**) PR flexible joint structural representation, No.: 1. flexible guiding pair; 2., 3., and 4. flexible hinges; 5. resistance arm; 6. flexible cantilever beam; 7. restricted column.

2.3.2. PR-Type Displacement Transfer Amplification Structure Design

Because the foot contact pressure is perpendicular to the ground, the resulting deformation is small and difficult to measure. To enable the flexible joint to capture small displacements, a PR-type transfer amplification structure is proposed. Among the known micro-displacement amplification structures, commonly used structures include the Scott–Russell structure [28,29], the bridge amplification structure, and the lever amplification structure [30].

The PR-type transfer amplification structure is based on the lever-type micro-displacement amplification structure and is composed of a PR (translation and rotation) flexible motion pair, which includes a parallel flexible guiding structure (moving pair) and a flexible hinge structure that does not produce over-constraints. The flexible motion pair has the advantages of no clearance, no mechanical friction, no wear, small size, and no need for lubrication, which can effectively ensure the real-time and reliable measurement of the flexible joint. Based on the requirements for a compact structure of the flexible joint, we made the input displacement and output displacement in opposite directions. As shown in Figure 5, the parallel flexible guiding structure that does not produce over-constraints can increase the structural stiffness in the vertical direction. The flexible hinge structure can transfer limited angular displacement around a fixed point, which depends on the characteristic of the location with thin thickness in the structure that can withstand certain elastic deformation and rebound in time [31].

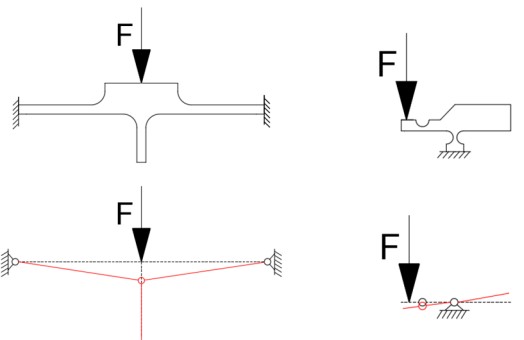

**Figure 5.** PR structure deformation schematic diagram.

During operation, the reaction force generated by the contact with the ground is transmitted to the input end of the flexible joint, which produces a small vertical displacement. This displacement is amplified by the lever structure with the flexible hinge as the fulcrum, and the lever end outputs a correspondingly amplified displacement. The lever-type micro-displacement amplification structure is connected to the pressure signal

output structure, which is the flexible cantilever beam structure. Through this structure, the output displacement is further amplified and then output.

The flexible hinge uses the small-angle elastic deformation of the thinnest part of its overall structure to achieve hinged-like motion and transmit force and displacement [32]. It combines the advantages of a flexible motion pair and has the advantages of high sensitivity and fast response. The structure is shown schematically in Figure 6. According to the different geometric structures, flexible hinges can be divided into two types: single-type and compound-type [33]. The single-type includes semi-circular notch and rectangular notch types, and the semi-circular-notch flexible hinge is the most basic and simplest, including parabolic, hyperbolic, single-axis double-circle, and single-axis single-circle types. Different types of flexible hinges correspond to different rigidities and transmission characteristics, and the selection should be based on the actual situation. According to the large flexibility required by commonly used multi-stage amplification structures and the requirements for torsional stiffness and rotation angle, the PR-type transfer amplification structure design uses commonly used single-axis double-circle and single-axis single-circle flexible hinges. These two flexible hinge structures have high precision and high rigidity and are easy to process. The torsional stiffness of the single-axis double-circle flexible hinge around the *z*-axis under the bending moment M(x) is [34]

$$k_z = \frac{2Ebt^{5/2}}{9\pi R^{1/2}} \tag{17}$$

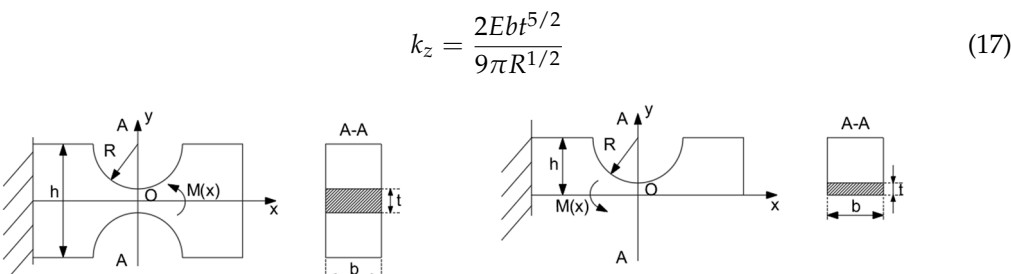

(**a**) Single-axis dual-circle flexible hinge    (**b**) Single-axis single-circle flexible hinge

**Figure 6.** Flexible hinge structure.

The torsional stiffness of a single-axis single-circle flexible hinge around the *z*-axis under the bending moment M(x) can be expressed by the following formula:

$$k_z = \frac{\sqrt{2}Ebt^{5/2}}{9\pi R^{1/2}} \tag{18}$$

In the formula:
$k_z$ represents the torsional stiffness;
$E$ represents the elastic modulus of the material;
$t$ represents the minimum thickness in the flexible hinge, and it is required that $t << R$;
$b$ represents the width of the flexible hinge;
$R$ represents the radius of the cutout in the flexible hinge.

From Formulas (16) and (17), it can be seen that the torsional stiffness of the flexible hinge is proportional to the minimum thickness and width, inversely proportional to the radius, and independent of the height. For the same size, the torsion angle of the single-axis double-circle flexible hinge will be larger under the same torque. Therefore, considering the overall stiffness of the PR-type micro-displacement amplification structure, the torsional stiffness of the flexible hinge itself, and ensuring the amplification factor, a single-axis single-circle flexible hinge with smaller torsional stiffness is added at the corner, and a single-axis double-circle flexible hinge is added at the fulcrum. The radius of the circular cutout in the flexible hinge is 0.75 mm.

### 2.3.3. Solution of PR-Type Flexible Joint Magnification

The effective amplification factor is a key parameter for evaluating the performance of the lever-type micro-displacement amplification structure and the flexible joint. Figure 7 shows the working principle of the flexible joint. The input displacement din is transmitted to the lever-type micro-displacement amplification structure through the parallel flexible guiding structure for the first-stage displacement amplification. Since the deformation of the parallel flexible guiding structure is small, it can be regarded as a rigid body in this calculation. Therefore, the input displacement applied to one end of the lever is

$$\Delta x_1 = d_{in} \tag{19}$$

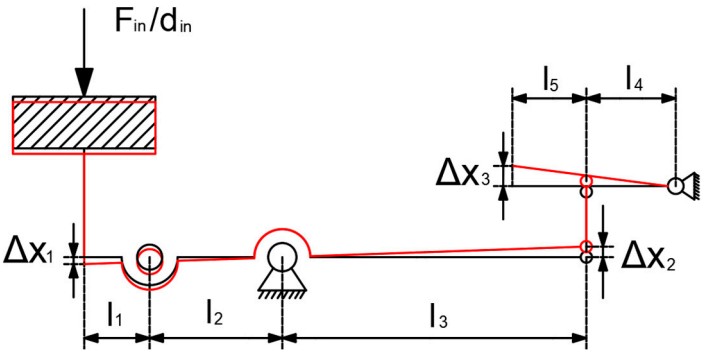

**Figure 7.** Working principle diagram of a flexible joint.

The lever-type micro-displacement amplification structure designed includes flexible hinges, and since there is midpoint drift in the flexible hinge, there is a gain angle $\Delta\theta$ at the flexible hinge. The output displacement $\Delta x_2$ at the end of the lever can be expressed as Formula (20):

$$\Delta x_2 = l_3 \cdot \tan(\theta + \Delta\theta) \tag{20}$$

$$\theta = \arctan \frac{\Delta x_1}{(l_1 + l_2)} \tag{21}$$

In Formula (21), $\theta$ represents the angle between the resistance arm and the undeformed lever when the gain angle is not considered.

The displacement of the lever-type micro-displacement amplification structure is directly applied to the flexible cantilever beam structure, and the output displacement Dout can be expressed as:

$$d_{out} = \frac{l_4 + l_5}{l_4} \Delta x_2 \tag{22}$$

Combining the above formulas, the effective magnification of the PR-type flexible joint is derived, as shown in Formula (23).

$$R_{amp} = \frac{d_{out}}{d_{in}} = \frac{(l_4 + l_5) \cdot tan \left( arctan \frac{\Delta x_1}{l_1 + l_2} + \Delta\theta \right)}{l_4 \cdot \Delta x_1} \tag{23}$$

### 2.4. Finite Element Simulation Analysis of Flexible Joints

In this section, in order to verify the accuracy and reliability of the static model of the flexible joint and optimize the key parameters in the joint, modeling and simulation analysis of the flexible joint are carried out, and the simulation results are compared with the static modeling results. In order to ensure structural rigidity, the main material of the flexible joint selected is 6061 alloy, the material of the pressure signal output structure selected is 65 Mn, and the main characteristic parameters are shown in Table 1.

**Table 1.** Flexible joint material parameters.

| Material | Parameters | | | |
| --- | --- | --- | --- | --- |
| | Elastic Modulus E (GPa) | Poisson's Ratio ν | Yield Strength σ (MPa) | Density ρ |
| 6061-T6 aluminum alloy (main part) | 68.9 | 0.33 | 275 | 2700 |
| 65 Mn (signal output structure) | 206 | 0.3 | 78.4 | 7.85 |

According to the structure size (Figure 8) of the PR sensor, the size parameters can be obtained, as shown in Table 2.

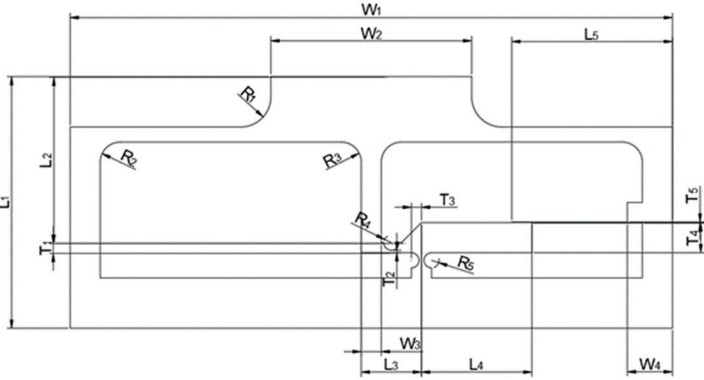

**Figure 8.** PR flexible joint size diagram.

**Table 2.** Size parameters of PR flexible joint.

| Parameters | Length/mm | Parameters | Length/mm |
| --- | --- | --- | --- |
| $L_1$ | 25 | $R_1$ | 2 |
| $L_2$ | 16.5 | $R_2$ | 2 |
| $L_3$ | 5 | $R_3$ | 2 |
| $L_4$ | 11 | $R_4$ | 0.75 |
| $L_5$ | 16 | $R_5$ | 0.75 |
| $T_1$ | 1 | $W_1$ | 60 |
| $T_2$ | 0.25 | $W_2$ | 20 |
| $T_3$ | 0.5 | $W_3$ | 2 |
| $T_4$ | 3 | $W_4$ | 4.5 |
| $T_5$ | 0.1 | Joint width | 60 |

2.4.1. Stiffness Finite Element Analysis of Flexible Joint Sensor

The process of obtaining the structural stiffness based on the finite element analysis results is as follows:

(1) Define the material properties of the structure according to the material parameters in the table. Then, fix the flexible joint with constraints and select the contact surface between the end of the lever-type micro-displacement amplification structure and the flexible cantilever beam as non-penetrating and the global contact as bonded. During operation, the flexible joint is subjected to a surface force that is uniformly distributed; therefore, a uniformly distributed surface force is applied at the input end with four spans.

(2) Obtain the simulation results and, based on the relationship between the input force Fin, output displacement Dout, and stiffness k, as shown in Equation (A41) (Appendix A), obtain the stiffness of the flexible joint.

The input–output simulation diagram of the flexible joint is shown in Figure 9. Based on the simulation results, the simulated stiffness value of the flexible joint is calculated. By substituting different input force values into the formula, the theoretical stiffness value is calculated and compared with the simulation results, as shown in Table 3. The relative error is within 8%, indicating that the theoretical model has a relatively high accuracy.

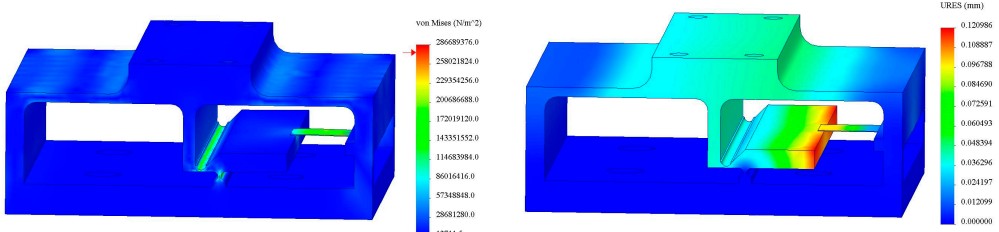

**Figure 9.** Finite element analysis of stiffness model of PR flexible joint sensor.

**Table 3.** Comparison of simulation stiffness and theoretical stiffness.

| Input Force $F_{in}$/N | Output Displacement $D_{out}$/μm | Stiffness k/(N/μm) | | Relative Error (%) |
| --- | --- | --- | --- | --- |
| | | Simulation Stiffness | Theoretical Stiffness | |
| 100 | 21.52 | 4.6468 | 5.0065 | 7.74 |
| 300 | 64.57 | 4.6461 | 5.0118 | 7.87 |
| 500 | 107.6 | 4.6468 | 5.0098 | 7.81 |
| 900 | 193.7 | 4.6464 | 5.0148 | 7.93 |

The reason for the differences between the theoretical model and the simulation results is that the transfer matrix method divides the entire structure into multiple modules for independent analysis and then connects the various modules based on force equilibrium relationships, ignoring the deformation between the modules. Additionally, the theoretical model is established under ideal conditions, ignoring detailed deformation.

Meanwhile, the output displacement of the flexible joint under different forces is significantly different. According to Section 2.3.1, when the maximum pushing force on the support leg is 689 N, the local maximum stress occurs at the flexible hinge, which is 205 Mpa, less than the yield strength of the 6061-T6 material, which is 275 Mpa. It can be seen that the flexible joint has good stability under this size parameter.

2.4.2. Finite Element Analysis of Effective Magnification Model of PR Flexible Joint

The process of obtaining the structural stiffness based on the finite element analysis results is as follows:

(1) Define the material properties of the structure according to the material parameters in the table. Then, fix the flexible joint with constraints and select the contact surface between the end of the lever-type micro-displacement amplification structure and the flexible cantilever beam as non-penetrating and the global contact as bonded. During operation, the flexible joint is subjected to a uniformly distributed surface force with an area of 1172 mm$^2$; therefore, different displacements of 5 μm, 10 μm, 20 μm, and 50 μm can be directly input to the surface to obtain the corresponding output displacement.

(2) Based on the simulation results, the output displacement of the feedback unit can be directly obtained, and the amplification factor can be calculated based on the input and output displacements. The input, output displacements, and amplification factors are shown in Tables 3 and 4.

**Table 4.** Comparison of simulation magnification and theoretical magnification.

| Input Displacement Din (μm) | Output Displacement Dout/μm | Magnification | | Relative Error (%) |
|---|---|---|---|---|
| | | Simulation Stiffness | Theoretical Stiffness | |
| 5 | 17.65 | 3.51 | 3.20 | 9.69 |
| 10 | 35.30 | 3.53 | 3.30 | 6.97 |
| 20 | 70.59 | 3.53 | 3.56 | 1.40 |
| 50 | 176.48 | 3.53 | 3.60 | 1.94 |

The simulation deformation results of the flexible joint are shown in Figure 10. Based on the simulation results, it can be seen that the effective amplification factor of the designed flexible joint and the effective amplification factor calculated theoretically have a relative error of less than 10% in the results of each input displacement. Additionally, the relative error decreases first and then increases as the input displacement increases. This proves the correctness of the theoretical model for the effective amplification factor.

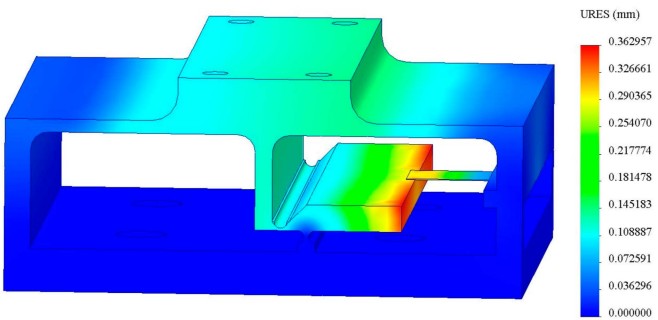

**Figure 10.** Finite element analysis of effective magnification model.

### 3. Results and Discussion

*3.1. Design of Performance Testing Experiments*

3.1.1. Performance Test Plan of PR Flexible Joint Sensor

To test the performance of the flexible joint sensor, the sensor was directly installed on the platform support leg for testing. In the static state of the platform, the flexible joint was compressed and released by raising and lowering the support leg, and the performance analysis was carried out by analyzing the output process data.

To verify the performance of the flexible joint in different usage scenarios, four sets of suspension tests were designed, each with different suspension sequences and holding times:

(1) The first set of test cycles was set to 22 s, with an initial state of grounding and holding for 10 s, then transitioning to a suspended state and holding for 2 s, and finally transitioning back to the grounding state and holding for 10 s.

(2) The second set of test cycles was set to 30 s, with an initial state of grounding and holding for 5.5 s, then transitioning to a suspended state and holding for 2.5 s, alternating between the grounding and suspended states twice, and finally transitioning back to the grounding state and holding for 6 s.

(3) The third set of test cycles was set to 20 s, with an initial state of suspension and holding for 5 s, then transitioning to the grounding state and holding for 10 s, and finally transitioning back to the suspended state and holding for 5 s.

(4) The fourth set of test cycles was set to 40 s, with an initial state of suspension and holding for 10 s, then transitioning to the grounding state and holding for 10 s, alternating between the suspension and grounding states once, and finally transitioning back to the suspension state and holding for 10 s.

To verify the performance of the flexible joint sensor, tests were conducted in the four sets of experiments. The flexible joint sensor was tested for response time in two working modes: suspension-grounding and grounding-suspension, with five repeated tests conducted for each mode. This provided data support for the precise control of the self-balancing control system.

3.1.2. Field Test Plan for Self-Balancing System of Mobile Robots in Greenhouse

To test the effectiveness of the self-balancing system, field tests were conducted on the mobile robot platform in a facility environment. The test environment is shown in Figure 11. Considering the possible road conditions that may be encountered in a greenhouse, the following road conditions were designed in the test, including climbing hills, traversing obstacles, and crossing furrows, as shown in Figure 12.

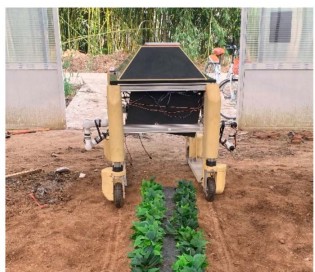

**Figure 11.** Field test environment of self-balancing mobile robot.

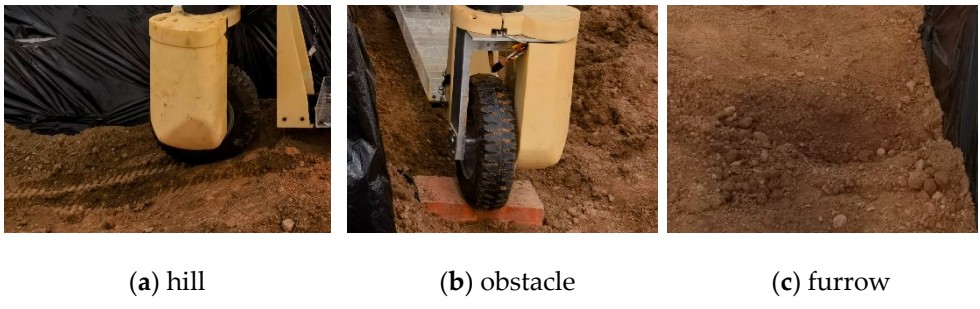

        **(a)** hill                   **(b)** obstacle               **(c)** furrow

**Figure 12.** Field test's road condition design.

During the experiment, a 1.5 m wide and 5 m long area was selected as the test area, and the platform was driven straight at a speed of 20 cm/s, passing through the above road conditions in sequence to reach the endpoint. The setting speed was 20 cm/s, mainly considering the movement speed required by the mobile platform when carrying a robotic arm to perform tasks such as picking, spraying, and fertilization. The inclination angle information was collected using a gyroscope during the process. After the experiment, the collected inclination angle information was processed, and the self-balancing results were analyzed. The experiments were conducted in the order of traversing obstacles, climbing hills, and crossing furrows, and each experiment was carried out in two states: with the self-balancing system turned on and with the self-balancing system turned off. The effectiveness of the self-balancing system was evaluated by comparing the results of the experiments.

(a)    The environment of climbing a hill

The field environment is complex, with significant terrain inclination and uneven ups and downs. To simulate the real field operation environment, a hill with a gradient of 0.18 was designed based on the most common terrain, with a hill below 0.2 in the actual environment. The hill was placed on the platform's travel path and passed through.

(b)    The environment of traversing obstacles

To simulate common obstacles in the field, such as rocks and field edges, a rectangular obstacle with a length of 120 mm, a width of 25 mm, and a height of 35 mm was designed.

(c)　　The environment of crossing a furrow

Another common terrain in the field environment is furrows, which are often used for drainage, irrigation, and other field operations. To simulate the furrow scenario, a furrow model with a width of 100 mm and a depth of 30 mm was designed.

### 3.2. Mobile Platform Robot Performance Test Results and Analysis

3.2.1. Performance Test Results and Analysis of PR Flexible Joint Sensor

According to the experimental plan in Section 3.1.1, tests were conducted on the flexible joint sensor, and the test results after taking the mean of three repeated tests are shown in Figure 13.

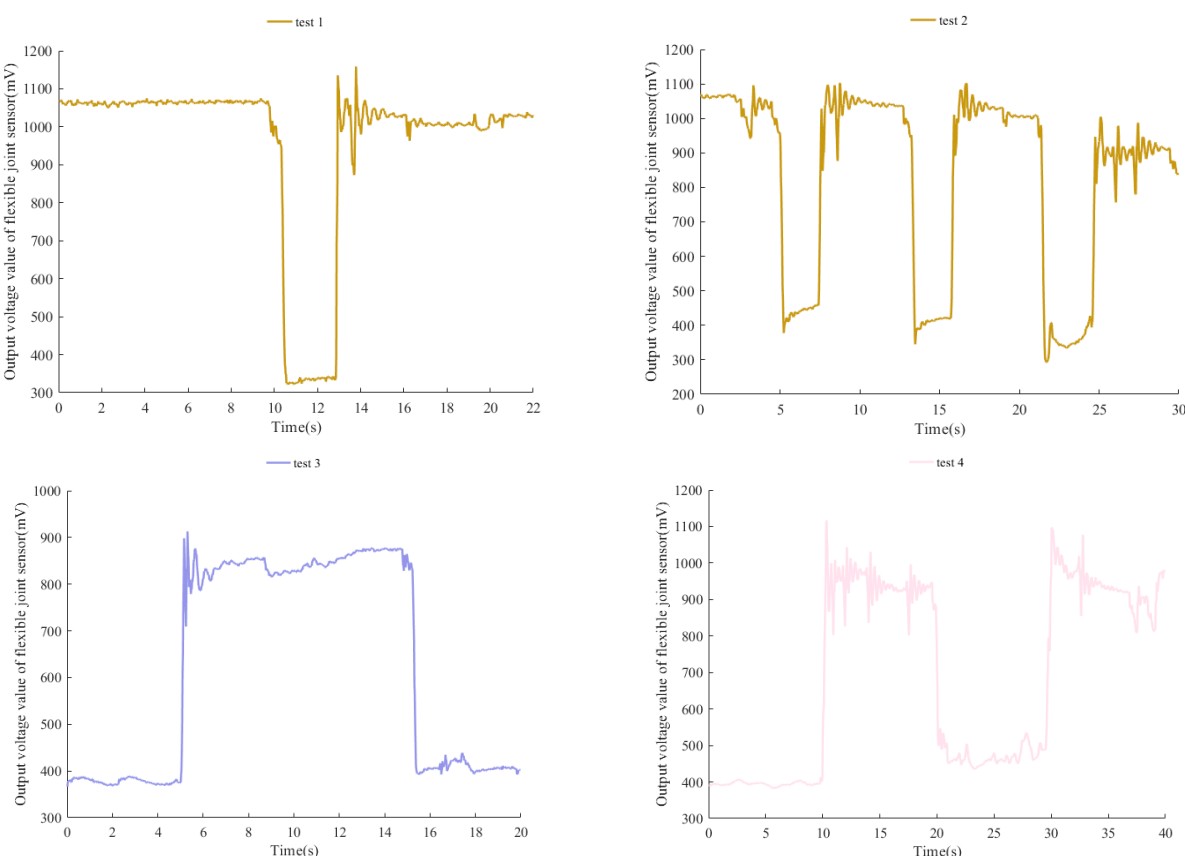

**Figure 13.** The change diagram of leg suspension voltage.

As shown in the figure, when the support leg is in a suspended state, the detection voltage of the flexible joint sensor is between 300 mV and 500 mV. When the support leg is in a grounded state, the detection voltage of the flexible joint sensor is between 900 mV and 1100 mV. The voltage change during the transition from grounding to suspension is about 500 mV, which can be detected by the controller and identified as a suspended state to solve the "weak leg" problem by providing detection information. The voltage fluctuation in the figure is relatively large, especially when the suspended state transitions to the grounded state; the voltage fluctuation is more obvious. This is mainly due to the overshoot and lag of the PID control algorithm used to control the leveling motor; therefore, mean filtering is added during detection to filter out noise interference.

Response time is one of the indicators that characterize sensor performance. Five sets of experiments were conducted in the suspension-grounding and grounding-suspension working modes, respectively, and the response time of the flexible joint sensor is shown

in Table 5. According to the data in the table, the mean response time of the flexible joint sensor is 0.16 s when the support leg transitions from a suspended state to a grounded state and 0.12 s when it transitions from a grounded state to a suspended state, indicating that the response speed meets the requirements for leveling.

**Table 5.** Flexible joint sensor's response time test results.

| Operational Mode | Response Time (s) | Average Response Time (s) |
|---|---|---|
| suspension-grounding | 0.11<br>0.16<br>0.21<br>0.15<br>0.18 | 0.16 |
| grounding-suspension | 0.14<br>0.09<br>0.10<br>0.14<br>0.13 | 0.12 |

### 3.2.2. Field Performance Test Results and Analysis of the Greenhouse Mobile Robot Self-Balancing System

According to the experimental plan in Section 3.1.2, the self-balancing system of the mobile robot was tested. The following are the test results after averaging the three repeated tests of the self-balancing system and the evaluation of the operating performance of the self-balancing system in the field test environment based on the test results. In the following figure, 'AL-on' represents turning on the self-balancing system, while 'AL-off' represents turning off the self-balancing system".

According to the inclination angle of the climbing test shown in Figure 14, when the self-balancing system is turned off, the maximum longitudinal inclination angle is 6.39°, the mean is 1.87°, and the standard deviation is 2.14°. The maximum lateral inclination angle is 4.92°, the mean is 1.65°, and the standard deviation is 1.71°. After the self-balancing system is turned on, the maximum longitudinal inclination angle is 1.42°, the mean is 0.41°, and the standard deviation is 0.47°. The maximum lateral inclination angle is 1.22°, the mean is 0.40°, and the standard deviation is 0.43°. The self-balancing system of the mobile robot can effectively reduce the longitudinal and lateral inclination angles.

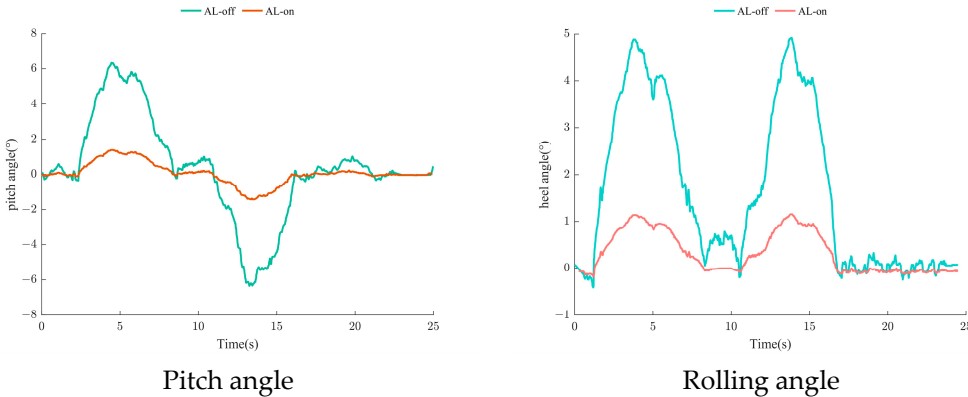

| Pitch angle | Rolling angle |

**Figure 14.** The angle of the hill climbing test.

Based on Table 6, when the self-balancing system is turned on, the maximum values of both the longitudinal and lateral inclination angles are reduced by an average of 76.49%, the mean values are reduced by an average of 76.92%, and the standard deviations are reduced by an average of 76.45%. The self-balancing system has a significant effect on adjusting the inclination angles under the climbing road conditions.

**Table 6.** The results of the hill climbing test.

| Hill Climbing Test | Extremal Value | | Mean Value | | Standard Deviation | |
|---|---|---|---|---|---|---|
| | Pitch Angle | Rolling Angle | Pitch Angle | Rolling Angle | Pitch Angle | Rolling Angle |
| AL-off | 6.39 | 4.92 | 1.87 | 1.65 | 2.14 | 1.71 |
| AL-on | 1.42 | 1.22 | 0.41 | 0.40 | 0.47 | 0.43 |
| reduce ratio | 77.78% | 75.20% | 78.07% | 75.76% | 78.04% | 74.85% |

Compared with the inclination angle data of climbing, the inclination angle data of the obstacle-crossing tests have a higher frequency of fluctuations. This is mainly because the height difference in the climbing environment gradually changes during the experiment, while the height difference in obstacle crossing changes vertically, which poses a greater challenge to the leveling system. According to Figure 15, when the self-balancing system is turned off, the maximum longitudinal inclination angle is 2.09°, the mean is 0.3°, and the standard deviation is 0.41°, while the maximum lateral inclination angle is 1.51°, the mean is 0.21°, and the standard deviation is 0.26°. After the self-balancing system is turned on, the maximum longitudinal inclination angle is 0.53°, the mean is 0.08°, and the standard deviation is 0.1°, while the maximum lateral inclination angle is 0.71°, the mean is 0.04°, and the standard deviation is 0.1°.

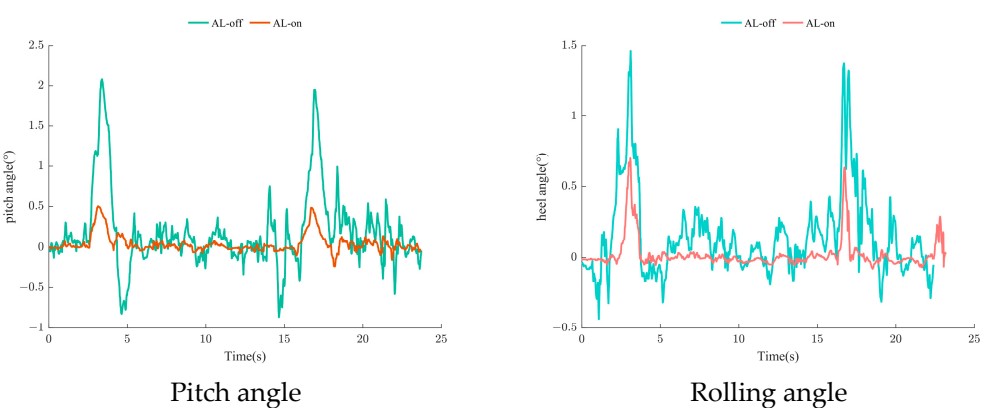

Pitch angle        Rolling angle

**Figure 15.** The angle of the traversing obstacle test.

Based on Table 7, when the self-balancing system is turned on, the maximum values of both the longitudinal and lateral inclination angles are reduced by an average of 63.81%, the mean values are reduced by an average of 77.14%, and the standard deviations are reduced by an average of 68.58%. The self-balancing system has a significant effect on adjusting the inclination angles when overcoming obstacles, and it can pass through obstacles more stably, reducing the shaking frequency of the robot platform.

**Table 7.** The results of the traversing obstacle test.

| Traversing Obstacle Test | Extremal Value | | Mean Value | | Standard Deviation | |
|---|---|---|---|---|---|---|
| | Pitch Angle | Rolling Angle | Pitch Angle | Rolling Angle | Pitch Angle | Rolling Angle |
| AL-off | 2.09 | 1.51 | 0.30 | 0.21 | 0.41 | 0.26 |
| AL-on | 0.53 | 0.71 | 0.08 | 0.04 | 0.10 | 0.10 |
| reduce ratio | 74.64% | 52.98% | 73.33% | 80.95% | 75.61% | 61.54% |

The main difference between the furrow test and the obstacle-crossing and climbing tests is that during the furrow test, either the front or rear wheels of the robot will enter the furrow. Due to the difference between the furrow test environment and the ideal environment, the two wheels of the robot cannot enter the furrow at the same time. This

leads to a significant difference in the longitudinal and lateral inclination angles when the front wheel enters the furrow compared to when the rear wheel enters the furrow, as shown in Figure 16. When the self-balancing system is turned off, the maximum longitudinal inclination angle is 1.79°, the mean is 0.24°, and the standard deviation is 0.40°, while the maximum lateral inclination angle is 2.15°, the mean is 0.24°, and the standard deviation is 0.40°. After the self-balancing system is turned on, the maximum longitudinal inclination angle is 0.35°, the mean is 0.08°, and the standard deviation is 0.09°, while the maximum lateral inclination angle is 0.44°, the mean is 0.04°, and the standard deviation is 0.07°.

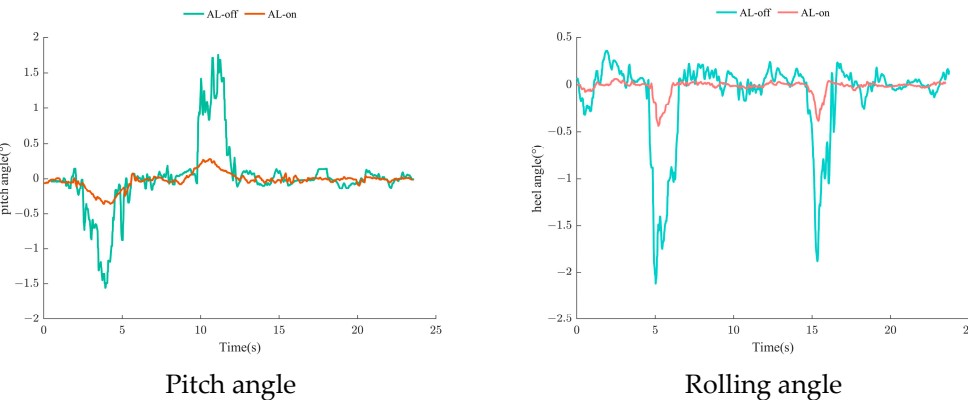

Pitch angle                                     Rolling angle

**Figure 16.** The angle of the crossing furrow test.

Based on Table 8, when the self-balancing system is turned on, the maximum values of both the longitudinal and lateral inclination angles are reduced by an average of 79.99%, the mean values are reduced by an average of 75%, and the standard deviations are reduced by an average of 80%. The self-balancing system has a significant effect on adjusting the inclination angles under the crossing ditch condition.

**Table 8.** The results of the crossing furrow test.

| Crossing Furrow Test | Extremal Value | | Mean Value | | Standard Deviation | |
|---|---|---|---|---|---|---|
| | Pitch Angle | Rolling Angle | Pitch Angle | Rolling Angle | Pitch Angle | Rolling Angle |
| AL-off | 1.79 | 2.15 | 0.24 | 0.24 | 0.40 | 0.40 |
| AL-on | 0.35 | 0.44 | 0.08 | 0.04 | 0.09 | 0.07 |
| reduce ratio | 80.45% | 79.53% | 66.67% | 83.33% | 77.5% | 82.5% |

Based on the data of the longitudinal and lateral inclination angles obtained in three walking environments, the self-balancing system of the mobile robot can perform effective posture adjustment and maintain the robot platform close to the horizontal position. In the field test environment, the adjusted longitudinal and lateral inclination angles were less than 1.5° less than 1.74° in the research of Fan Guiju et al. [11]. The comprehensive leveling performance of this study is superior to the solution proposed by Fan Guiju et al.

In general, the field application scenarios and the characteristics of the objects being adjusted can affect the leveling performance of the leveling system. For example, the performance of the leveling system for agricultural machinery may vary in scenarios such as gentle slopes and mountainous terrains.

## 4. Conclusions

This study has developed a self-balancing mobile robot suitable for narrow greenhouse environments, which provides a reliable and stable moving platform for information collection, precise fertilization, precise spraying, and picking systems. The greenhouse self-balancing mobile robot reduces the impact of undulating terrain on the system's accuracy and operation efficiency. The three-degrees-of-freedom wheel-leg design of the robot

ensures its flexibility in walking. A self-balancing system combining a gyroscope and a PR flexible joint sensor has been designed and developed, which can autonomously adjust the posture of the robot platform. Simulation experiments on the PR flexible joint sensor have verified its rigidity and micro-displacement amplification effect, which are consistent with theoretical expectations.

In addition, performance tests of the flexible joint sensor and field performance tests of the self-balancing system were conducted in this study. The experimental results show that the voltage change between the ground-falling and hanging states of the supporting legs is about 500 mV, and the average response time of hanging-falling and falling-hanging is 0.16 s and 0.12 s, respectively. The experimental results show that the flexible joint sensor has a good real-time characterization of whether the foot is in good contact with the ground, and it solves the problem of "virtual legs". Self-balancing-system comparative tests were conducted in three terrain scenarios. The maximum inclination angles in the obstacle-crossing, climbing, and ditch-crossing scenarios decreased by 63.81%, 76.49%, and 79.99%, respectively, when the self-balancing system was turned on compared to when it was turned off, while the mean inclination angles decreased by 77.14%, 76.92%, and 75%, respectively, and the inclination angle variances decreased by 68.58%, 76.45%, and 80%, respectively, indicating a significant improvement in the self-balancing system performance.

The greenhouse self-balancing mobile robot proposes a self-balancing system based on PR sensors. This self-balancing system introduces a new approach for small-scale mobile robots to acquire foot pressure, providing a fresh solution to address the issue of "weak legs" in mobile robots.

**Author Contributions:** Conceptualization, Y.Z., Y.S. and K.Z.; methodology, Y.Z. and F.L.; validation, D.Z., L.Y., X.H. and T.C.; formal analysis, Y.Z., Y.S. and F.L.; investigation, F.L. resources, K.Z.; data curation, Y.S.; writing—original draft preparation, Y.Z.; writing—review and editing, D.Z., K.Z., L.Y., X.H. and T.C.; visualization, Y.Z. and Y.S.; supervision, K.Z.; project administration, K.Z.; funding acquisition, K.Z. All authors have read and agreed to the published version of the manuscript.

**Funding:** This research received no external funding.

**Institutional Review Board Statement:** Not applicable.

**Data Availability Statement:** The data used in this study are self-tested and self-collected. As the control method designed in this paper is still being further improved, the data cannot be shared at present.

**Conflicts of Interest:** The authors declare no conflict of interest.

## Appendix A

(1)　　Transfer matrix calculation method

In this paper, the flexible joint uses a lever-type micro-displacement amplification structure; therefore, the beam element is taken as an example to derive its transfer matrix. The transfer matrix represents the transfer relationship between the force and displacement at both ends of the beam element, and the transfer direction is shown in Figure A1, that is, the relationship between the A-end nodal force $F_a$, the displacement $D_a$ and the B-end nodal force $F_b$ and the displacement $D_b$. Among them, the nodal force F includes the axial force $F_x$, the shear force $F_y$, and the bending moment M. Displacement D includes lateral displacement $D_x$, longitudinal displacement $D_y$, and corner α.

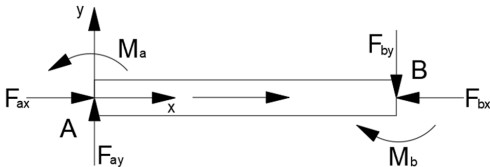

**Figure A1.** *Cont.*

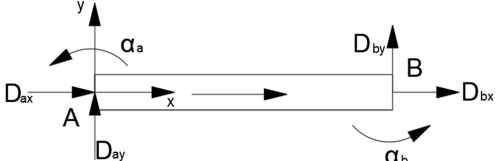

**Figure A1.** Flexible beam element.

The transfer relation of the beam element from end A to end B is:

$$\begin{bmatrix} F_B \\ D_B \end{bmatrix} = T \begin{bmatrix} F_A \\ D_A \end{bmatrix} \tag{A1}$$

In the formula: $F_A = [F_{ax}\ F_{ay}\ M_a]^T$, $D_A = [D_{ax}\ D_{ay}\ \alpha a]^T$, $F_B = [F_{bx}\ F_{by}\ M_b]^T$, $D_B = [D_{bx}\ D_{by}\ \alpha_b]^T$.

T is the transfer matrix, and the formula for solving the transfer matrix according to the principle of force balance and virtual work is:

$$T = \begin{bmatrix} 1 & 0 & 0 & 0 & 0 & 0 \\ 0 & 1 & 0 & 0 & 0 & 0 \\ 0 & -L & 1 & 0 & 0 & 0 \\ 0 & 0 & 0 & & & \\ t_{41} & 0 & 0 & 1 & 0 & 0 \\ 0 & t_{52} & t_{53} & 0 & 1 & L \\ 0 & t_{62} & t_{63} & 0 & 0 & 1 \end{bmatrix} \tag{A2}$$

In the formula:
$t_{41} = -\int_0^L \frac{dx}{EA(x)}$; $t_{52} = \int_0^L \frac{(L-x)xdx}{EI(x)} - \int_0^L \frac{udx}{GA(x)}$; $t_{53} = -\int_0^L \frac{(L-x)dx}{EI(x)}$; $t_{62} = \int_0^L \frac{xdx}{EI(x)}$;
$t_{63} = -\int_0^L \frac{dx}{EI(x)}$;
L represents the beam element length;
E represents the modulus of elasticity;
A(x) represents the cross-sectional area function;
I(x) represents the inertia product function;
G represents shear modulus;
u represents the stress distribution unevenness coefficient.

The transfer matrix in the independent module coordinate system can be obtained from Formula (A2). When the overall recursive model is considered, and the node force balance is established, it is necessary to unify the transformation of each independent module into the overall structural coordinate system.

As shown in Figure A2, when the beam unit AB rotates β counterclockwise around point A, the nodal force and bending moment of A and B in the independent coordinate system $Ox'y'$ are $F'_{ax}$, $F'_{ay}$, $M'_a$, $F'_{bx}$, $F'_{by}$, and $M'_b$. In the global coordinate system Oxy, the nodal force and bending moment of A and B are $F_{ax}$, $F_{ay}$, $M_a$, $F_{bx}$, $F_{by}$, and $M_b$. The following transformation relations exist between the two coordinate systems:

$$F'_A = RF_A, D'_A = RD_A, F'_B = RF_B, D'_B = RD_B \tag{A3}$$

In the formula, R is the rotation matrix, which is defined as:

$$R = \begin{bmatrix} \cos\beta & \sin\beta & 0 \\ -\sin\beta & \cos\beta & 0 \\ 0 & 0 & 1 \end{bmatrix} \tag{A4}$$

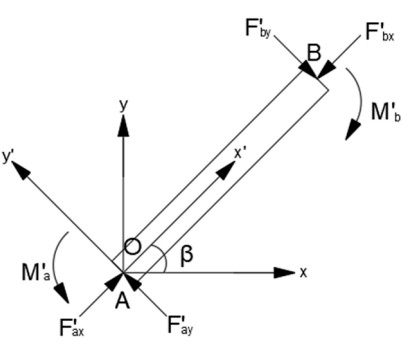
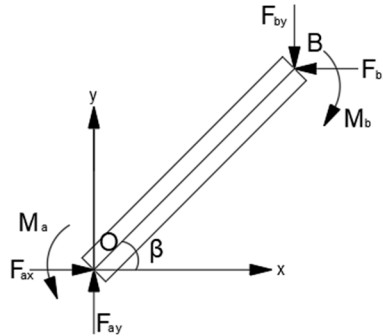

(**a**) Independent coordinate system      (**b**) Global coordinate system

**Figure A2.** Unified overall structure coordinate system.

From Formula (A1), it can be seen that the transfer relationship from end A to end B of the beam element in the independent coordinate system is:

$$\begin{bmatrix} F\prime_B \\ D\prime_B \end{bmatrix} = T\prime \begin{bmatrix} F\prime_A \\ D\prime_A \end{bmatrix} \tag{A5}$$

Substituting Equation (A3) into Equation (A5) provides:

$$\begin{bmatrix} F_B \\ D_B \end{bmatrix} = \lambda^T T\prime \lambda \begin{bmatrix} F_A \\ D_A \end{bmatrix} \tag{A6}$$

In the formula:

$$\lambda = \begin{bmatrix} R & 0 \\ 0 & R \end{bmatrix}$$

The transfer matrix of the beam element in the global coordinate system can be obtained as $T = \lambda^T T\prime \lambda$.

(2)    Stiffness solution of flexible guide pair module

In this paper, the structural size and transmission direction of the flexible guiding sub-module in the flexible joint in the independent coordinate system are shown in Figure A3.

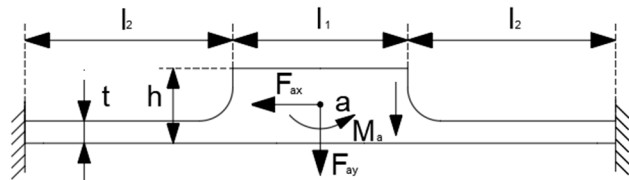

**Figure A3.** Flexible guide pair (Module 1): $\prime l_1 \prime$ boss length, $\prime l_2 \prime$ cantilever beam length, 't' cantilever beam thickness, and 'h' boss height.

The relationship between force and displacement at the flexible guiding auxiliary node a is:

$$F_a = K_a D_a \tag{A7}$$

Among them, $K_a$ is the stiffness matrix of the flexible guide pair, which is defined as

$$K_a = \begin{bmatrix} \frac{2Ebt}{l_2} & 0 & 0 \\ 0 & \frac{2Ebt^3(1-\mu^2)}{l_2^3} & 0 \\ 0 & 0 & \frac{2Ebt^3}{3l_2} \end{bmatrix} \tag{A8}$$

Among them, b represents the length of the flexible guide pair in the *z*-axis direction, and $\mu$ denotes the material Poisson's ratio.

(3) Solution of transfer matrix of flexible hinge module

The structural size and transmission direction of the flexible hinge module in the independent coordinate system are shown in Figure A4.

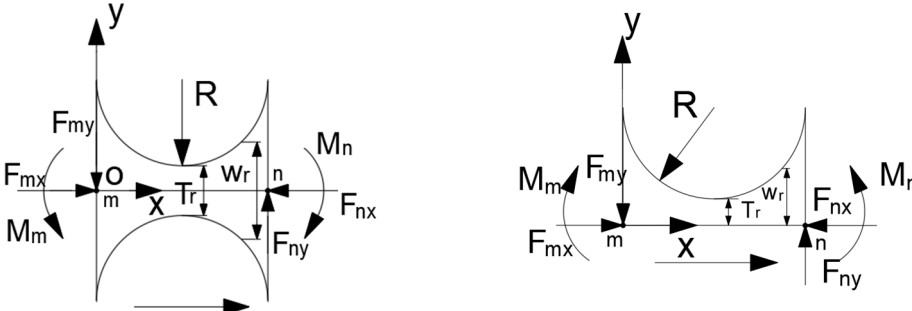

**Figure A4.** Independent coordinate system model of single-axis double-circle and single-axis single-circle flexible hinge module: /$T_r$/ minimum thickness, /$W_r$/ thickness, and /$R$/ radius.

The transmission relationship between the force and displacement of the flexible hinge from the left node m to the right node n is:

$$\begin{bmatrix} F_n \\ D_n \end{bmatrix} = T_c \begin{bmatrix} F_m \\ D_m \end{bmatrix} \tag{A9}$$

The flexible hinge can be regarded as a beam whose cross-section changes regularly, and its transfer matrix can be obtained by direct integration. Among them, the variable thickness $W_r$ of the single-axis double-circle flexible hinge is:

$$W_r(x) = 2R + T_r - 2\sqrt{R^2 - (x - R)^2} \tag{A10}$$

The variable thickness $W_r$ of the single-axis single-circle flexible hinge is:

$$W_r(x) = R + T_r - \sqrt{R^2 - (x - R)^2} \tag{A11}$$

Substitute Formulas (A10) and (A11) into Formula (A2) to obtain the transfer matrices $T_{c1}$ and $T_{c2}$. Taking the single-axis single-circle flexible hinge as an example, let $z = \frac{R}{T_r}$, the elements in $T_{c2}$ are as follows:

$$t_{41}^r = -\frac{1}{Eb}\left[\frac{(2z+1)\sqrt{2}}{\sqrt{4z+1}}\arctan\sqrt{4z+1} - \frac{\pi}{4}\sqrt{2}\right] \tag{A12}$$

$$t_{52}^r = -\frac{12}{Eb}\left[\frac{\sqrt{2\left(36z^5+28z^4+15z^3+4z^2+z\right)}}{2(2z+1)(4z+1)^2}\right.$$
$$+ \frac{\sqrt{2}(2z+1)\left(36z^4+4z^3-7z^2+4z-1\right)}{2(4z+1)^{\frac{5}{2}}}$$
$$\left. - \frac{u}{Gb}\left[\frac{\sqrt{2}(2z+1)}{\sqrt{4z+1}}\arctan\sqrt{4z+1} - \frac{\pi}{4}\sqrt{2}\right]\right] \tag{A13}$$

$$t_{53}^r = -\frac{24z^2}{Ebt_r}\left[\frac{3\sqrt{2}\left(2z^2+z\right)}{2(4z+1)^{\frac{5}{2}}}\arctan\sqrt{4z+1} + \frac{3z^2+2z+1}{(4z+1)^2(2z+1)}\right] \tag{A14}$$

$$t_{62}^r = -\frac{24z^2}{Ebt_r}\left[\frac{3\sqrt{2}(2z^2+z)}{(4z+1)^{\frac{5}{2}}}\arctan\sqrt{4z+1} + \frac{3z^2+2z+1}{(4z+1)^2(2z+1)}\right] \qquad (A15)$$

$$t_{63}^r = -\frac{24z^2}{Ebt_r}\left[\frac{3\sqrt{2}(2z^2+z)}{(4z+1)^{\frac{5}{2}}}\arctan\sqrt{4z+1} + \frac{3z^2+2z+1}{(4z+1)^2(2z+1)}\right] \qquad (A16)$$

Modules 3 and 4 in the flexible joint are flexible hinges, and module 3 is a single-axis single-circle flexible hinge. According to the transformation rules of the global coordinate system and the transmission direction shown in the figure, the transfer matrix is $T_3 = T_{c2}$.

Module 4 is a single-axis double-circle flexible hinge, and its transfer matrix is:

$$T_4 = \lambda(-90°)^T T_{c1}(R_3, T_3)\lambda(-90°)^T \qquad (A17)$$

(4)   Solution of transfer matrix of flexible beam module

The structural size and transmission direction of the flexible beam module in the independent coordinate system are shown in Figure A5.

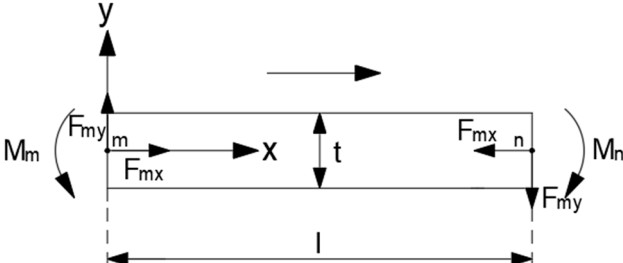

**Figure A5.** Flexible beam module /l/, Length, and /t/ Thickness.

The transfer relationship between the force and displacement of the flexible beam from the left node m to the right node n is:

$$\begin{bmatrix} F_n \\ D_n \end{bmatrix} = T_l\begin{bmatrix} F_m \\ D_m \end{bmatrix} \qquad (A18)$$

Substitute the size parameters into Formula (A2) to obtain the transfer matrix $T_l$ of the flexible beam module. Among them, the element value of the solution is:

$$t_{41}^l = -\frac{l}{Ebt} \qquad (A19)$$

$$t_{52}^l = -\frac{2l^3}{Ebt^3} - \frac{ub}{GBt} \qquad (A20)$$

$$t_{53}^l = -\frac{6l^2}{Ebt^3} \qquad (A21)$$

$$t_{62}^l = \frac{6l^2}{Ebt^3} \qquad (A22)$$

$$t_{63}^l = -\frac{12l}{Ebt^3} \qquad (A23)$$

Modules 2, 5, and 6 in the flexible joint can be approximately regarded as flexible beams, and then, according to the transformation rules of the global coordinate system and the transmission direction shown in the figure, the transfer matrix is:

$$T_2 = \lambda(-90°)^T T_1(l_3, t_3)\lambda(-90°)^T \tag{A24}$$

$$T_5 = T_1(l_5, t_5) \tag{A25}$$

$$T_6 = \lambda(180°)^T T_1(l_6, t_6)\lambda(180°)^T \tag{A26}$$

According to the above analysis, the transfer functions of each module have been obtained, and the stiffness of the overall structure can be solved through the force–transfer relationship. To analyze the force–transfer relationship, firstly, carry out a force analysis on each module, and the force analyses of modules 2, 3, 4, 5, and 6 are shown in Figure A6.

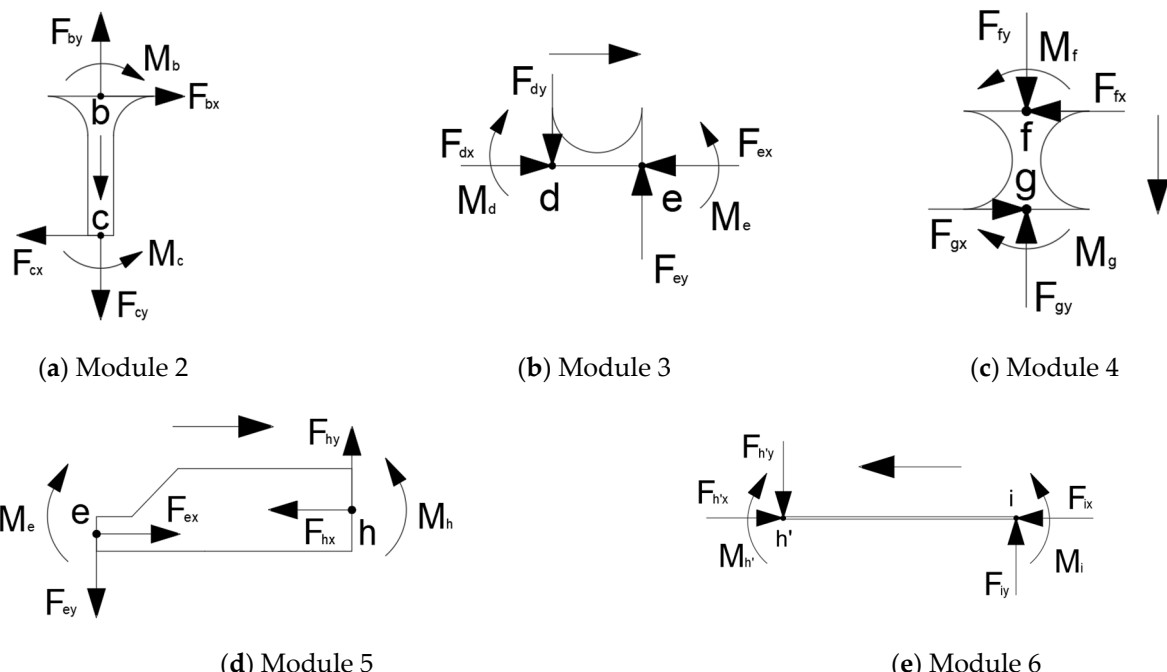

(**a**) Module 2  (**b**) Module 3  (**c**) Module 4

(**d**) Module 5  (**e**) Module 6

**Figure A6.** Force analysis of each module.

According to the force analysis and transmission relationship, the relationship between the force and displacement between nodes is established:

$$\begin{bmatrix} F_c \\ D_c \end{bmatrix} = T_2 \begin{bmatrix} F_b \\ D_b \end{bmatrix} \tag{A27}$$

$$\begin{bmatrix} F_e \\ D_e \end{bmatrix} = T_3 \begin{bmatrix} F_d \\ D_d \end{bmatrix} \tag{A28}$$

$$\begin{bmatrix} F_g \\ D_g \end{bmatrix} = T_4 \begin{bmatrix} F_f \\ D_f \end{bmatrix} \tag{A29}$$

$$\begin{bmatrix} F_h \\ D_h \end{bmatrix} = T_5 \begin{bmatrix} F_e \\ D_e \end{bmatrix} \tag{A30}$$

$$\begin{bmatrix} F_h \\ D_h \end{bmatrix} = T_6 \begin{bmatrix} F_i \\ D_i \end{bmatrix} \tag{A31}$$

According to the principle of force analysis, it can be known that points c and d point force, e point force, e' point force, h point force, and h' point force act on each other; then, we can obtain:

$$|F_c| = |F_d|, |D_c| = |D_d| \tag{A32}$$

$$|F_e| = |F_{e'}|, |D_e| = |D_{e'}| \tag{A33}$$

$$|F_h| = |F_{h'}|, |D_h| = |D_{h'}| \tag{A34}$$

For the uniaxial double-circle flexible hinge of module 4 as the fulcrum of the enlarged lever, its force and displacement on point *f* are:

$$F_f = F_{e'} - F_h \tag{A35}$$

$$D_f = D_e \tag{A36}$$

$$\begin{bmatrix} F_f \\ D_f \end{bmatrix} = \begin{bmatrix} F_{e'} \\ D_{e'} \end{bmatrix} - \begin{bmatrix} F_h \\ 0 \end{bmatrix} \tag{A37}$$

In addition, since one end of it is fixed, its stiffness is set as K_s, then the relationship between its force and displacement can be expressed as

$$F_f = K_s D_f \tag{A38}$$

At this time, assuming that the input force is $F_{in} = [0 \ F_y \ 0]^T$, then acting on the flexible joint can be expressed as

$$F_b = F_a + F_{in} \tag{A39}$$

According to the above relationship, it can be obtained that:

$$\begin{bmatrix} F_h \\ D_h \end{bmatrix} = T_6 T_5 T_3 T_2 \begin{bmatrix} K_a K_s & 0 \\ 0 & K_a K_s \end{bmatrix} \tag{A40}$$

$$F_{in} = K D_{h'} \tag{A41}$$

In the formula, K represents the stiffness matrix of the flexible joint, and its inverse matrix is the flexibility matrix of the flexible joint; therefore, the output displacement $D_{h'}$ and the input force Fin can be expressed as

$$D_{h'} = C F_{in} \tag{A42}$$

Assuming that the stiffness of the flexible joint is k, it can be deduced that the relationship between the input force $F_y$ of the flexible guide pair and the output displacement $D_{h'}$ of the flexible cantilever beam is:

$$F_y = k D_{h'} \tag{A43}$$

According to the above derivation, the flexibility matrix C can be expressed as

$$C = \begin{bmatrix} c_{11} & c_{12} & c_{13} \\ c_{21} & c_{22} & c_{23} \\ c_{31} & c_{32} & c_{33} \end{bmatrix} \tag{A44}$$

From this, the flexible joint stiffness k can be deduced in the formula.

$$k = \frac{1}{c_{22}}$$

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
