# Peer review of "Design and Experiment of Greenhouse Self-Balancing Mobile Robot Based on PR Joint Sensor"

_agriculture, doi:10.3390/agriculture13102040_

Round 1

Reviewer 1 Report

Comments and Suggestions for Authors

1.The introduction discusses the current research on active balance mobile platforms, but it fails to highlight the significance of the PR sensor-based active leveling solution in this study. Please provide a description of this solution's importance in the introduction.

2.The workflow and principles of the active leveling solution based on PR sensors are lacking in the section "2. Research on Design and Research of Greenhouse Self-balancing Mobile Robot." Please provide a clear description to supplement this section.

3.The advantages of using PR sensors for detecting the contact force between wheels and the ground are not adequately described in section "2.3. The design of PR flexible joint sensor " Please provide a clear explanation to address this.

4.In section "2.1 Design scheme and working principle of self-balancing mobile robot in greenhouse," the roles of different parts in the design of the greenhouse self-balancing mobile robot (as shown in Figure 1) lack relevant descriptions. Please provide additional explanations to clarify this.

5.In section "2.1 Design scheme and working principle of self-balancing mobile robot in greenhouse " the roles of different components in the design of the three-degree-of-freedom leg (as shown in Figure 2) lack relevant descriptions. Please supplement this section with appropriate explanations.

6.The conclusion and outlook section lacks a comprehensive summary of the contributions of this research to active balance mobile platforms. Please provide a summary description to address this point.

Author Response

We appreciate the modifications suggested by the reviewer for our manuscript. The word file "author-coverletter-32452565.v3" is the modifications we have made in response to the suggestions. 

Reviewer 2 Report

Comments and Suggestions for Authors

Dear authors,

Please find below my comments about your article. I found the presented work very interesting, however, I think some improvements must be made.

The abstract describes a study of a greenhouse self-balancing mobile robot to avoid issues such as the inability of the greenhouse working robot to perform normal tasks or reduced working accuracy due to the influence of uneven ground. The self-balancing mobile robot system uses a quadruped mobile robot as a carrier, equipped with a three-degree-of-freedom wheel-leg structure, and matched with a posture control algorithm. The algorithm calculates the adjustment of each leg based on the vehicle’s tilt angle and wheel-ground pressure, achieving control over the robot’s posture angle.

This paper claims that the presented system is a less complex solution than other multi-degree-of-freedom leg designs while still being suitable for greenhouse environments. To address the issue of (weak legs) posture adjustment during mobile robot fieldwork, a flexible joint sensor based on the PR structure is presented.

General comments:

  • The introduction must be improved.  The authors mention that one of the main contributions of this study is the design of the flexible joint sensor to solve the (weak legs) problem. However, very little is explained in the introduction about how other systems try to solve or solve this issue in order to compare then against the proposed solution.
  • An explanation of the type of sensor they are using as PR sensor must be included. There is a quite large section explaining the PR-type transfer amplification structure, but I couldn't find any note about the sensor you are using.
  • I think something very interesting to know about the reported tests is the speed at which the robot moves linearly while preforming them. Which is the maximum speed at which the robot can move while the system is still capable keeping the platform straight?
  • The authors must perform more tests to get more statistically valid measurements.
  • The authors must  present some analysis on how their results perform when compared to some of the works cited in the introduction to make the scientific contribution clearer.
  • Please make an extensive review of grammar and syntax errors of the whole document.

Specific comments

  • (47-51) It was not clear what you wanted to say here. I think there are some incomplete sentences or you wanted to say above rather than below. Please check this paragraph.
  • (96) The "ghost legs" problem is not explained, I assumed that it is the same as the (weak leg) problem but you state this clearly. I also had troubles finding the explanation about what do you mean by the (weak leg) problem, maybe you should explain this earlier in your document or cite the section where you explain this the first time you use the term. 
  • (151) possible incomplete sentence please verify it.
  • (149-150) It is hard to understand how the PR flexible sensor is installed it would be great if you could  add a close-up image of how this looks in reality. Maybe you could add it as a detail in figure 2 trying to relate it with your Figure 4.
  • 197 Space missing "Punder"
  • 227 and 229 , wa and wb symbols don't match to your equation (14).
  • 232 commas are strange.
  • 233-237 You explained the (weak leg problem) here. However, it was not very clear (at least for me who had never heard the term before) what you mean by this. Please improve your explanation and/or add some reference to other sources where the problem is explained. I know this could be quite obvious when you are dealing with this problem daily but this is not the case for everybody.
  • 381 You say that you show your strawberry picking robot in figure 9. I think this is a mistake, please verify it.
  • 383-384 Figure number might be wrong.
  • 452 Describe the sensor.
  • 541  You should perform more tests.

Author Response

We appreciate the modifications suggested by the reviewer for our manuscript. The word file "author-coverletter-32452565.v1" is the modifications we have made in response to the suggestions. 

Round 2

Reviewer 2 Report

Comments and Suggestions for Authors

Dear authors

The text has been improved for this second version.

However, there are still some issues to address.

General comments

  •  I insist that the introduction section must be improved. The authors mention that one of the main contributions of this study is the design  of the flexible joint sensor to solve the (weak legs) problem. However, very little is  explained in the introduction about  how other systems try to solve or solve this issue in order to compare then against the proposed solution.
  • You  answered to one of my previous comments the following: The classification and description of other methods for obtaining foot pressure information are provided in 7 section 2.1.
  • Up to my best understanding Section 2.1 describes your own work. You don't mention other similar approaches or at least they are not being cited. More importantly, you are not comparing other  similar approaches to your work, therefore, your contribution on this field is not clear.

Specific changes

(45) Paragraph is incomplete. It was not in the previous version.

(254) As in this second version you have already explained the "weak leg" problem before this point. I think this line needs to be modified.

(264 - 270)  You could relate this paragraph to your figures 2 or 4 to improve clarity.

(408) There are some not-english characters in table 2.

(501) Explain  why during the field tests the robot speed was set to 20 cm/s. Was it just because it was the best-case scenario? Explain what happens to your system at different speeds.  Mention which are the limit  values at which your system no longer performs correctly.

(625-629)  In this paragraph you say that your study is superior to the solution proposed by Fan Guiju et al. But you don't explain how this is superior.

Author Response

Thank you to the experts for their suggestions on the revisions to our research. The following Word file provides answers to the revisions.
